

# The preclinical pharmacological study on HX0969W, a novel water-soluble pro-drug of propofol, in rats

YuJun Zhang[1,2,*], YingYing Jiang[1,*], HaiYan Wang[3], Bin Wang[4], Jun Yang[1,2], Yi Kang[1,2], Jun Chen[5], Jin Liu[1,2] and Wen-sheng Zhang[1,2]

[1] Department of Anesthesiology, Laboratory of Anesthesia and Critical Care Medicine, Translational Neuroscience Centre, West China Hospital, Sichuan University, Chengdu, China
[2] National-Local Joint Engineering Research Center of Translational Medicine of Anesthesiology, West China Hospital, Sichuan University, Chengdu, China
[3] Department of Anaesthesiology, Yuebei People's Hospital, Guangdong University, Shaoguan, China
[4] Department of Anesthesiology, Guizhou Provincial People's Hospital, Guiyang, China
[5] Laboratory Department Guizhou Provincial Corps Hospital of PAPF, Guiyang, China
* These authors contributed equally to this work.

Corresponding author
Wen-sheng Zhang,
zhang_ws@scu.edu.cn

## ABSTRACT

**Background**. Propofol is the most widely used intravenous sedative-hypnotic anesthetic in clinical practice. However, many serious side effects have been related to its lipid emulsion formulation. The pro-drug design approach was used to develop the water-soluble propofol, which could effectively resolve the limitations associated with the lipid emulsion formulation. Thus, the new water-soluble pro-drug of propofol, HX0969W, was designed and synthesized. The objective of this study was to conduct preclinical pharmacological studies on this novel water-soluble pro-drug of propofol.

**Methods**. The assessment of the loss of the righting reflex (LoRR) was used for the pharmacodynamic study, and liquid chromatography-tandem mass spectrometry and high-performance liquid chromatography- fluorescence were used for the pharmacokinetic study.

**Results**. The potency of HX0969W ($ED_{50}$ [95% CI], 46.49 [43.89–49.29] mg/kg) was similar to that of fospropofol disodium (43.66 [43.57–43.75] mg/kg), but was lower than that of propofol (4.82 [4.8–14.82] mg/kg). Administered with a dose of 2-fold $ED_{50}$, propofol required a shorter time to cause LoRR than that of HX0969W and fospropofol. However, the LoRR duration was significantly longer in response to the administration of HX0969W and fospropofol disodium than that caused by propofol. In the pharmacokinetic study, the $C_{max}$ of fospropofol was higher than that of HX0969W. HX0969W had a shorter mean residual time and a rapid clearance rate than that of fospropofol disodium. There was no significant difference between the $T_{max}$ of the propofol whether it was released by HX0969W or fospropofol disodium; the $C_{max}$ of propofol released by HX0969W was similar to that of propofol, which was higher than the propofol released by fospropofol disodium.

## INTRODUCTION

General anesthesia is a combination of drugs, including sedative-hypnotic agents, analgesics, or muscle relaxants, which put the patients in a sleep-like state before surgery or other medical examinations. Propofol is the most widely used intravenous sedative-hypnotic anesthetic in clinical practice (*Hemphill et al., 2019*). However, many serious side effects have been related to its lipid emulsion formations, such as emulsion instability, injection pain, hyperlipidemia, infection, fat metabolism disorder, and propofol-related infusion syndrome (*Diaz et al., 2014*; *Lee et al., 2014*; *Mirrakhimov et al., 2015*; *Pestana, Garcia-de Lorenzo & Madero, 1996*; *Prankerd & Stella, 1990*; *Singh, Jindal & Singh, 2011*; *Wachowski et al., 1999*; *Wolf et al., 2001*; *Zhou et al., 2015*). The pro-drug design approach was widely used to develop the novel water-soluble propofol, which could effectively avoid the limitations associated with its lipid emulsion formulation (*Feng et al., 2017*).

Fospropofol disodium is the first water-soluble pro-drug of propofol that has been approved by the U.S. Food and Drug Administration to be administered as a bolus injection for adult patients undergoing diagnostic or therapeutic procedures (*Moore, Walker & MacLaren, 2009*; *Telletxea et al., 2012*). Although it is confirmed that formaldehyde, which is one of the metabolites of fospropofol disodium, does not accumulate after a single administration (*Garnock-Jones & Scott, 2010*; *Kumpulainen et al., 2008*), it is still considered a potentially risk for systemic toxicity and has limited the clinical indication of fospropofol disodium for continuous infusion. A new water-soluble pro-drug of propofol was designed and synthesized in our laboratory (patent number, WO2011160268; denoted as HX0969W), which was metabolized to propofol, $\gamma$-hydroxybutyrate (GHB), and phosphate *in vivo*. GHB was found to be further converted into carbon dioxide and water (*Lang et al., 2014*; *Rousseau et al., 2012*). In contrast to fospropofol disodium, HX0969W cloud produce an effective sedative-hypnotic effect with lesser potential risks of systemic toxicity.

In the present study, we evaluated the median effective dose ($ED_{50}$) of HX0969W, fospropofol disodium, and propofol for the loss of righting reflex (LoRR) in rats using the up and down method. Then, we measured the onset time and duration for HX0969W, fospropofol disodium and propofol after single intravenous injections. Furthermore, the pharmacokinetic parameters of the drugs were assessed in rats.

## MATERIAL AND METHODS

### Materials

HX0969W and fospropofol disodium were synthesized at Yichang Humanwell Pharmaceutical Co., Ltd. (Yichang, China). Propofol was purchased from AstraZeneca (Shanghai, China). 7-Hydroxycoumarin, thymol, and ammonium acetate were purchased from Sigma-Aldrich Co., Ltd. (MO, USA). Acetonitrile and methanol were obtained from ROE Scientific Inc. (DE, USA). Ultrapure water was produced using Milli-Q® integral water purification system (Merck Millipore, Germany).

## Experimental animals

Adult Sprague-Dawley rats (age: 8–10 weeks, body weight: 220–350 g) were purchased from Dossy Biological Technology Co., Ltd. (Chengdu, China) and housed in polypropylene cages (less than 5 animals per cage) at the Experimental Animal Center of Sichuan University (Chengdu, China) at an ambient temperature of $25 \pm 1\ °C$, a controlled humidity of 50%–70%, and a 12-h light-dark cycle (7 a.m. to 7 p.m.). The rats were provided with a radiation-sterilized commercial diet and filtered water *ad libitum*. The rats were subjected to fasting for 12 h with uncontrolled water supply prior to initiation of dosing; food was supplied immediately after dosing. The experimental interventions and sample collections in this study were accomplished through the tail vein after venipuncture (Terumo® intravenous catheter; 24G, Tokyo, Japan). Thus, no anesthetic procedure was used in this study. At the end of experiments, all rats were euthanized *via* overdose of pentobarbital sodium. All experiments were performed with the approval of the Committee of Scientific Research and the Institutional Animal Experimental Ethics Committee of West China Hospital, Sichuan University, Chengdu, China (2015015A).

## Measurement of the hypnotic median effective dose

The $ED_{50}$ values of HX0969W, fospropofol disodium, and propofol were determined by the up and down method (*Dixon, 1991*; *Glen & Hunter, 1984*; *Kilpatrick et al., 2007*). The tail vein was cannulated with a Terumo® intravenous catheter (24G; Tokyo, Japan) for drug administration. After administered the drug, the rats were assessed one by one in separate box for the LoRR effect (once the rat lay prone and no reaction to mild stimuli, the rat was gently changed to the supine position recognized as LoRR), and were provided with warm blanket and oxygen inhalation to reduce external stimulation. If the duration of LoRR after drug administration was more than 30 s, the next rat was administered with a lower dose. On the contrary, if the duration of LoRR was less than 30 s (none-LoRR), a higher dose was injected in the next rat. Recording the LoRR to none-LoRR as a cross, the measurement of the $ED_{50}$ value was terminated once more than five crosses occurred. An accurate $ED_{50}$ value with 95% confidence intervals (CI) was calculated using standard computing equations (*Dixon, 1991*; *Glen & Hunter, 1984*; *Kilpatrick et al., 2007*).

## Pharmacodynamic study in rats

Thirty adult Sprague-Dawley rats were randomly divided into three groups to evaluate the efficacy of HX0969W, fospropofol disodium, and propofol ($n = 10$ in each group, 5 males and 5 females). The drugs were administered *via* tail vein through the Terumo® intravenous catheter (24G; Tokyo, Japan) at equivalent doses (2-fold dose of $ED_{50}$ for LoRR in rats) at an injection speed of 0.25 mL/s. Then, each rat was placed in a separate box with a warm blanket and oxygen inhalation. For the pharmacodynamic study, the time to LoRR (onset time) and time to recovery from LoRR (duration) were recorded. All rats were observed closely for any mortality, behavioural changes, and clinical symptoms of toxicity.

## Pharmacokinetic study in rats

Before the experiment, the rats were subjected to fasting for 12 h with uncontrolled water supply; food was supplied immediately after drug administration. The tail vein was cannulated with a Terumo® intravenous catheter (24G; Tokyo, Japan) for drug administration and sample collection. HX0969W, fospropofol disodium, and propofol were injected at their equivalent doses in the rats through their tail veins ($n = 10$ in each group, 5 males and 5 females). The blood samples from the rats were collected at different times, which were determined based on the results of the preliminary experiments. For HX0969W and fospropofol disodium, 50 µL blood samples were collected at 1, 2, 3, 4, 5, 7, 10, 15, 20, 30, 45, 60, 90, 120, 180, and 240 min after drug administration. For propofol, 50 µL blood samples were collected at 0.5, 1, 2, 3, 5, 7, 10, 15, 20, 30, 45, 60, 90, 120, and 180 min after drug administration.

### *Sample preparation and quantitative method*

Blood samples from the HX0969W and fospropofol disodium groups were deproteinized by methanol solution with 7-Hydroxycoumarin (internal standard, IS). The mixture was centrifuged for 10 min at 25,000 × g at 4 °C after vortexing. The supernatant was then subjected to determination and quantification by liquid chromatography-tandem mass spectrometry (LC-MS/MS). The LC-MS/MS analysis system consisted of an Agilent 6460 triple quadrupole mass spectrometer (Agilent Technologies, CA, USA) with an electrospray ionization source. Chromatographic separation was performed using a Zorbax eclipse plus C8 column (100 mm × 2.1 mm, 3 µm) at 12 °C with 5 mM ammonium acetate in deionized water and acetonitrile at a volume ratio of 65: 35 for HX0969W and 70: 30 for fospropofol disodium, at a flow rate of 0.3 mL/min. The mass spectrometry conditions were as follows: negative ionization mode; the sheath gas flow rate, 5.0 l/min; sheath gas heater temperature, 350 ° C; nebulizer pressure, 30 psi; capillary voltage, 4,500 V; *m/z* 343.0 →177.0 for HX0969W, *m/z* 287.00 →177.00 for fospropofol disodium, and *m/z* 160.7 →89.0 for IS.

For propofol, the samples were deproteinized with a methanol solution with thymol (IS). The mixture was centrifuged for 10 min at 25,000 × g at 4 °C after vortexing. The supernatant was then subjected to determination and quantification by high-performance liquid chromatography- fluorescence. The analysis system consisted of an Agilent Zorbax XDB (Agilent Technologies, CA, USA) equipped with a C18 column (150× 4.6 mm, 5 µm) and a fluorescence detector. The liquid chromatography conditions were as follows: mobile phase solvent, acetonitrile and water at the volume ratio of 40: 60 at a flow rate of 1.2 mL/min; fluorescence detector, the wavelength of excitation and emission at 276 and 310 nm, respectively.

The method validation results were detailed in the supplementary material includes the following parameters: specificity, linearity, lower limit of quantitation (LLOQ), precision, accuracy, matrix effect, extraction recovery, stability and dilution integrity.

### *Pharmacokinetic analysis*

Noncompartmental pharmacokinetic methods were used for pharmacokinetic analysis of HX0969W, fospropofol disodium, and propofol using the Phoenix Winnonlin® software

(version 6.3, NJ, USA). The pharmacokinetic parameters used in this study were as follows: maximum concentration ($C_{max}$) and the time to acquire ($T_{max}$), area under curve (AUC) from zero to the last time point ($AUC_{0-t}$), mean residual time (MRT), half-time ($t_{1/2}$), and clearance (CL).

### Data analysis

The statistical analyses were performed using SPSS (version 21, IL, USA). The $ED_{50}$, presented as mean and 95% CI, was calculated using the up and down method. The Kruskal–Wallis test followed by the Mann–Whitney $U$ test was used for the data that did not fit normality, and the one-way *ANOVA* followed by Tukey's test was used for the data that did not fit normality and equality of variance. The level of statistical significance was set at $p < 0.05$.

## RESULTS

### Measurement of the median effective dose for the sedative-hypnotic effect

Following a single intravenous administration of the drug, each rat was placed in a separate box to record the time to LoRR and time to recovery from LoRR. Then, the median effective dose to achieve the sedative-hypnotic effect (Table 1) was calculated using the up and down method (*Dixon, 1991*; *Glen & Hunter, 1984*; *Kilpatrick et al., 2007*). The potency of HX0969W in rats ($ED_{50}$ [95% CI], 46.49 [43.89–49.29] mg/kg) was found to be similar to that of fospropofol disodium ($ED_{50}$ [95% CI], 43.66 [43.57–43.75] mg/kg), but lower than that of propofol ($ED_{50}$ [95% CI], 4.82 [4.81–4.82] mg/kg). Due to the pro-drug design, HX0969W and fospropofol disodium are metabolized to propofol, which produces the sedative-hypnotic effect. Thus, the dosages of HX0969W and fospropofol used for general anesthesia were higher than those of propofol.

### Pharmacodynamic study in rats

After administering a 2-fold $ED_{50}$ dose for LoRR, the time to LoRR (onset time) and time to recovery from LoRR (duration) were measured in the rats. No mortality or significant clinical symptoms of toxicity were observed in these rats. As for the onset time (Fig. 1A), propofol ($0.4 \pm 0.1$ min) required a shorter time to cause LoRR than that required for HX0969W ($1.8 \pm 0.4$ min, $p = 0.017$) and fospropofol disodium ($2.1 \pm 0.7$ min, $p = 0.006$). However, the duration of LoRR for HX0969W ($75.8 \pm 9.6$ min, $p = 0.036$) and fospropofol disodium ($68.5 \pm 18.4$ min, $p = 0.041$) were significantly longer than that of propofol ($27.1 \pm 6.0$ min, Fig. 1B).

### Pharmacokinetic study of HX0969W and fospropofol disodium in rats

After a single intravenous injection, there was no significant difference in the action onset time and duration between HX0969W and fospropofol disodium. The efficacy of a drug is closely related to its pharmacokinetic features. Therefore, we measured the concentration–time profiles of HX0969W and fospropofol disodium in rats. The concentration–time curves are presented in Fig. 2, and the calculated pharmacokinetic parameters are shown in Table 2. Following the administration of equivalent doses (92.98 mg/kg for HX0969W, 87.32

**Table 1  Calculation process of ED$_{50}$ for HX0969W, fospropofol disodium and propofol.**

| Substance | Dose (mg/Kg) | X | Loss of the righting reflex (±) | t | C | M |
|---|---|---|---|---|---|---|
| HX0969W | 59.17 | 1.772 | + | 1 | 1.772 | 3.140 |
| | 53.25 | 1.726 | + + − + | 4 | 6.904 | 11.916 |
| | 47.93 | 1.681 | + + − + − − − | 7 | 11.767 | 19.780 |
| | 43.13 | 1.635 | − + − + + − | 6 | 9.810 | 26.682 |
| | 38.82 | 1.589 | − − − | 3 | 4.767 | 7.575 |
| | | | ED$_{50}$ = lg$^{-1}$ (ΣC/ Σt) = 46.49 mg/Kg (95%CI [43.894–49.293] mg/Kg) | | | |
| Fospropofol disodium | 61.80 | 1.791 | | 0 | 0 | 0 |
| | 55.62 | 1.745 | | 0 | 0 | 0 |
| | 50.06 | 1.700 | + + | 2 | 3.400 | 5.780 |
| | 45.05 | 1.654 | + − + + + + + + | 8 | 13.232 | 21.886 |
| | 40.55 | 1.607 | − − − − − − − | 7 | 11.249 | 18.077 |
| | | | ED$_{50}$ = lg$^{-1}$ (ΣC/ Σt) = 43.66 mg/Kg (95%CI [43.567–43.752] mg/Kg) | | | |
| Propofol | 6.00 | 0.778 | + + | 2 | 1.566 | 1.211 |
| | 5.04 | 0.702 | − + + + + − + + + | 8 | 5.616 | 3.942 |
| | 4.23 | 0.626 | − − − − − − − | 6 | 3.756 | 2.351 |
| | 3.56 | 0 | | 0 | 0 | 0 |
| | 2.99 | 0 | | 0 | 0 | 0 |
| | | | ED$_{50}$ = lg$^{-1}$ (ΣC/ Σt) = 4.82 mg/Kg (95%CI [4.818–4.821] mg/Kg) | | | |

**Notes.**

1, X = lg(dose); t = sum total of rat; C = X × t; M = $X^2$ × t. 2, + at with loss of the righting reflex; −: rat without loss of the righting reflex. 3,3, 95%CI = lg$^1$ (lgED$_{50}$ ± 1.96 slgED$_{50}$); SlgED$_{50}$ = [ΣM − (ΣC)$^2$/Σt]/(Σt(Σt − 1))1/2.

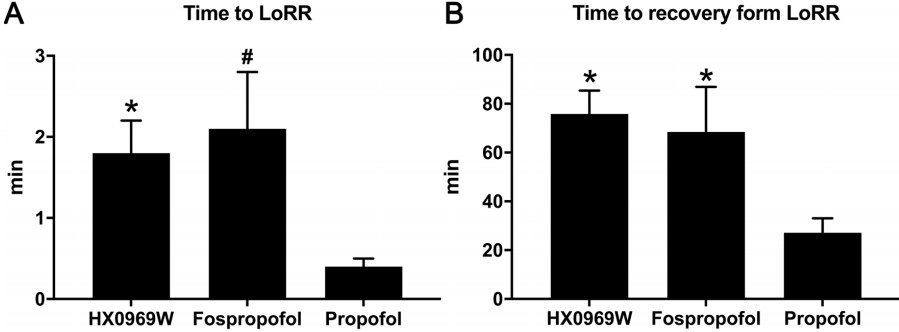

**Figure 1  The onset time (A) and duration (B) of hypnotic behavior observation after intravenous administration of 2-fold ED$_{50}$ drugs ($n = 10$ in each group).** Data are expressed as mean ± SD.*$p < 0.05$; # $p < 0.01$.

mg/kg for fospropofol disodium), the maximal concentration of fospropofol disodium (603.49 ± 411.29 µg/mL) was found to be higher than that of HX0969W (321.30 ± 67.22 µg/mL). In rat plasma, HX0969W had a shorter mean residual time (3.67 ± 1.71 min) and a rapid clearance rate (89.97 ± 15.94 ml/min/kg) than those of fospropofol disodium (13.15 ± 5.45 min and 31.12 ± 19.09 ml/min/kg, respectively). The computed AUC value of HX0969W from zero to the last time point was 1053.78 ± 214.37 min* µg/ml, which was less than that of fospropofol disodium (3804.92 ± 2091.75 min* µg/ml). Thus, we
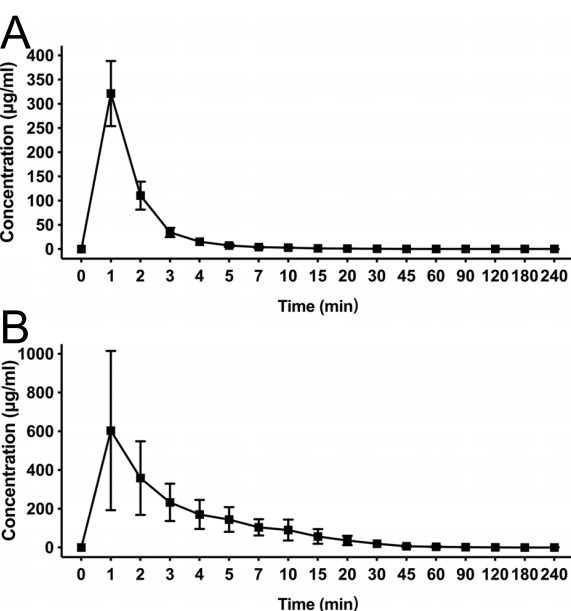

**Figure 2** **Mean concentration-time profiles of HX0969W (A) and fospropofol disodium (B) in plasma after intravenous administration with of 2-fold ED$_{50}$ in rats ($n = 10$ in each group).** Data are expressed as mean ± SD.

**Table 2** **Pharmacokinetic parameters of HX0969W and fospropofol (mean ± SD).**

| Parameters | HX0969W | fospropofol |
|---|---|---|
| C$_{max}$ (µg/ml) | 321.3 ± 67.22 | 603.49 ± 411.29 |
| t$_{1/2}$(min) | 71.79 ± 59.52 | 49.29 ± 40.31 |
| AUC$_{0\sim t}$ (min* µg/ml) | 1053.78 ± 214.37 | 3804.92 ± 2091.75 |
| MRT (min) | 3.67 ± 1.71 | 13.15 ± 5.45 |
| CL (ml/min/kg) | 89.97 ± 15.94 | 31.12 ± 19.09 |

**Notes.**
AUC, area under the curve]; MRT, mean residual time; CL, clearance.

considered that HX0969W, after a bolus injection in rats, had a faster metabolism *in vivo* as compared to fospropofol disodium. The pharmacokinetic parameters of HX0969W and fospropofol disodium could not explain the lack of significant differences in the onset time and duration. Therefore, we further evaluated the concentration of propofol released by HX0969W and fospropofol disodium in rats.

## Pharmacokinetic study of propofol in rats

Using an equivalent dose of HX0969W, fospropofol disodium and the parent drug propofol, the concentration–time curves are presented in Fig. 3, and the calculated pharmacokinetic parameters are shown in Table 3. The T$_{max}$ of propofol released by HX0969W (propofol$_H$, 4.0 ± 0.47 min) or fospropofol disodium (propofol$_F$, 4.5 ± 1.18 min) in plasma had no significant difference, which was longer than that for propofol (0.5 min). However, the C$_{max}$ of propofol$_H$ (24.26 ± 5.14 µg/ml) was similar to the parent drug propofol

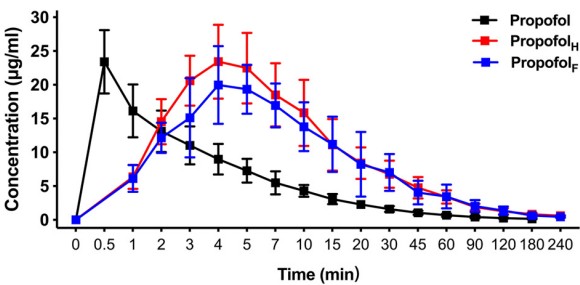

**Figure 3** Mean concentration-time profiles of propofol in plasma after intravenous administration of HX0969W, fospropofol disodium, and propofol at dose of 2-fold $ED_{50}$ in rats ($n = 10$ in each group). Data are expressed as mean ± SD. $Propofol_H$, propofol released by HX0969W; $Propofol_F$, propofol released by fospropofol disodium.

**Table 3** Pharmacokinetic parameters of propofol (mean ± SD).

| Parameters | Propofol | Propofol$_H$ | Propofol$_F$ |
|---|---|---|---|
| $C_{max}$ (µg/ml) | 23.4 ± 4.69 | 24.26 ± 5.14 | 21.91 ± 4.98 |
| $T_{max}$ (min) | 0.5 | 4.0 ± 0.47 | 4.5 ± 1.18 |
| $t_{1/2}$ (min) | 54.24 ± 16.18 | 88.62 ± 77.35 | 78.97 ± 29.05 |
| $AUC_{0\sim t}$ (min*µg/ml) | 219.23 ± 42.21 | 740.5 ± 186.33 | 704.53 ± 226.25 |
| MRT (min) | 29.45 ± 4.63 | 52.6 ± 2.90 | 50.92 ± 4.40 |
| CL (ml/min/kg) | 43.28 ± 8.46 | 54.67 ± 13.83 | 68.12 ± 22.84 |

**Notes.**

[a] Propofol$_H$, propofol released by HX0969W; Propofol$_F$, propofol released by fospropofol.

[b] AUC, area under the concentration; MRT, mean residual time; CL, clearance.

(23.4 ± 4.69 µg/ml), which was higher than that for propofol$_F$ (21.91 ± 4.98 µg/ml). The AUC, MRT, and CL of propofol$_F$ (740.5 ± 186.33 min* µg/ml, 52.6 ± 2.90 min, and 54.67 ± 13.83 ml/min/kg, respectively) were similar to those of propofol$_H$ (704.53 ± 226.25 min* µg/ml, 50.92 ± 4.40 min, and 68.12 ± 22.84 ml/min/kg, respectively), but were higher than those of propofol (219.23 ± 42.21 min* µg/ml, 29.45 ± 4.63 min, and 43.28 ± 8.46 ml/min/kg, respectively). In this pharmacokinetic study, the longer onset time and duration of HX0969W and fospropofol disodium, than those of the parent drug propofol, were due to their pharmacokinetic features, which included a larger AUC and a longer MRT. Furthermore, the onset time and duration of HX0969W and fospropofol disodium, which showed no significant difference, could be related to the undifferentiated pharmacokinetic features of propofol$_H$ and propofol$_F$, including $T_{max}$, $C_{max}$, AUC, MRT and CL.

## DISCUSSION

In this study, we have demonstrated that HX0969W requires a longer time to induce the sedative-hypnotic effect, but with longer duration than propofol in rats. HX0969W had a large $C_{max}$, a short MRT, and a rapid CL than fospropofol disodium in rats. The $T_{max}$ of propofol released by HX0969W or fospropofol disodium showed no significant difference,

and the $C_{max}$ of propofol released by HX0969W was similar to that of propofol, which was higher than that of the propofol released by fospropofol disodium.

The main objective of this study was to develop a new water-soluble pro-drug of propofol with non-toxic metabolites. In the pharmacodynamic study, the potency of HX0969W was found to be similar to that of fospropofol disodium, but lower than that of the parent drug propofol. Similar to fospropofol disodium, the onset time and duration of HX0969W after single intravenous administration, were longer than those of propofol. Thus, HX0969W is the new water-soluble pro-drug of propofol that can be used for induction and maintenance of general anesthesia. In contrast to fospropofol disodium, which is metabolized into a toxic metabolite, HX0969W was metabolized to propofol, GHB, and phosphate *in vivo*; GHB is further converted into carbon dioxide and water (*Lang et al., 2014*; *Rousseau et al., 2012*). Previous studies have demonstrated that GHB is mainly used as a sedative drug for children suffering from burns or for the treatment of alcohol and opioid addiction (*Brambilla et al., 2012*; *Gallimberti et al., 1993*; *Rousseau et al., 2012*). However, it was reported that GHB produced an obvious hypnotic effect after oral administration at least 800 mg/kg (*Lettieri & Fung, 1979*). In our published study, HX0969W produced a sedative-hypnotic effect after oral administration with 193.08 mg/kg in rats (*Wang et al., 2015*), which was far less than 800 mg/kg. Meanwhile, there is no data to show if there is any synergy between the effects of propofol and GHB. Therefore, the LoRR caused by HX0969W administration was mainly caused by the propofol released by HX0969W. As compared to fospropofol disodium, HX0969W cloud be an effective sedative-hypnotic agent with lesser potential risks of systemic toxicity.

In the open-label, single-arm, phase 3 clinical trial (*Gan et al., 2010*), five (4.1%) patients administered with fospropofol experienced the adverse events: hypoxemia ($n = 1$, 0.8%), hypotension ($n = 4$, 3.3%), bradycardia ($n = 1$, 0.8%). No patient experienced apnea during the procedure. In a randomized, double-blind, phase 3 study (*Cohen et al., 2010*), hypotension occurred in two patients (1.3%) in the fospropofol group, and one patient (0.6%) experienced hypoxemia that resolved after repeated verbal stimulation. Undergoing flexible bronchoscopy (*Silvestri et al., 2009*), the most common cardiopulmonary adverse events were hypoxemia ($n = 36$, 14.3%) and hypotension ($n = 8$, 3.2%). In another clinical trial (*Cohen, 2008*), four patients in fospropofol group experienced sedation-related adverse events: mild hypotension ($n = 2$, 2%) and hypoxemia ($n = 2$, 2%). Only one patient of hypoxemia required the airway assistance (verbal stimulation). Thus, fospropofol showed lower incidences of hypotension, respiratory depression, apnea, and loss of airway patency because of its slower onset of action (*Mahajan, Mahajan & Kaushal, 2012*). On the basis of the findings in this study, there is no significant difference between HX0969W and fospropofol disodium in pharmacodynamic and pharmacokinetic studies. Therefore, it would be reasonable to consider the possibility that HX0969W could reduce the cardiopulmonary side effects by a slow-released propofol as fospropofol disodium. In the subsequent experiments, we will focus on evaluating the safety of HX0969W and propofol in the cardiovascular and respiratory systems.

Propofol has been the most widely used intravenous anesthetic in clinical practices due to its rapid onset and recovery from sedation (*Hemphill et al., 2019*). However, the

pharmacokinetic features of HX0969W, including a longer MRT and $t_{1/2}$, and a larger AUC, have superseded the advantages of propofol. Therefore, HX0969W is more appropriate for patients undergoing longstanding surgeries or a long-term sedation in intensive care unit, and its use can avoid the side effects of propofol by its lipid emulsion preparation. Meanwhile, HX0969W generated a hypnotic effect with rapid onset and shorter duration than fospropofol disodium and propofol after oral administration in rats (*Wang et al., 2015*). Therefore, there might be a new clinical indication for HX0969W for use in patients undergoing pre-operative sedation, transitory diagnostic or therapeutic procedures by oral administration. Above all, our results show that HX0969W can avoid the disadvantages associated with the lipid emulsion formulation of propofol. Thus, HX0969W is more suitable for long-term sedation and pre-operative preparation for children who do not cooperate with venipuncture.

## CONCLUSIONS

With the equivalent dose, HX0969W and fospropofol disodium had a longer time to cause LoRR with longer duration than propofol in rats. In pharmacokinetic study, the $C_{max}$ of fospropofol disodium was higher than HX0969W. HX0969W had a shorter MRT and a rapid CL than fospropofol disodium. Due to the pro-drug design, HX0969W and fospropofol disodium are metabolized to propofol, which produces the sedative-hypnotic effect. The $T_{max}$ of propofol released by HX0969W and fospropofol disodium had no significant difference, which was longer than that for propofol. The $C_{max}$ of propofol released by HX0969W was similar to the parent drug propofol, which was higher than that for that of propofol released by fospropofol disodium.

## ACKNOWLEDGEMENTS

The authors would like to thank Linghui Yang and Deying Gong (Laboratory of Anaesthesia and Critical Care Medicine, Translational Neuroscience Centre, West China Hospital, Sichuan University, Chengdu, China) for the assistance in reviewing the manuscript. Meanwhile, we are also grateful to LinQiao Tang and Yan Wang from the Core facility of West China Hospital for their technical supports.

### Funding
The authors received no funding for this work.

### Competing Interests
WenSheng Zhang, Jun Yang, and Jin Liu are the inventors of the HX0969W patent (WO2011160268).

## Author Contributions

- YuJun Zhang and YingYing Jiang conceived and designed the experiments, analyzed the data, prepared figures and/or tables, authored or reviewed drafts of the paper, and approved the final draft.
- HaiYan Wang conceived and designed the experiments, performed the experiments, analyzed the data, prepared figures and/or tables, and approved the final draft.
- Bin Wang and Jun Chen analyzed the data, prepared figures and/or tables, and approved the final draft.
- Jun Yang conceived and designed the experiments, prepared figures and/or tables, and approved the final draft.
- Yi Kang performed the experiments, authored or reviewed drafts of the paper, and approved the final draft.
- Jin Liu and Wen-sheng Zhang conceived and designed the experiments, authored or reviewed drafts of the paper, and approved the final draft.

## Animal Ethics

The following information was supplied relating to ethical approvals (i.e., approving body and any reference numbers):

The Committee of Scientific Research and the Institutional Animal Experimental Ethics Committee of West China Hospital, Sichuan University, Chengdu, China approved this study (2015015A).

## Patent Disclosures

The following patent dependencies were disclosed by the authors:

Phosphoric acid ester compound of hydroxy acid substituted phenyl ester, synthesis method and medical use thereof, WO2011160268.

## Data Availability

The raw data are available in the Supplemental File.

## Supplemental Information

Supplemental information for this article can be found online at http://dx.doi.org/10.7717/peerj.8922#supplemental-information.

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
