# Peer review of "The preclinical pharmacological study on HX0969W, a novel water-soluble pro-drug of propofol, in rats"

_PeerJ, doi:10.7717/peerj.8922_

## Round 0.1 · original submission · Major Revisions

There are grammar errors throughout the manuscript (For examples, the sentences in line 156-157 and line 192-193 etc). Please correct them carefully in the revision version. In addition, you must perform additional experiments to compare the effect of HX0969W and propofol according to the comments from the reviewers.

Reviewer 1 ·

Basic reporting

Authors already reported the acitivities of HX0969W,
1) Sichuan Da Xue Xue Bao Yi Xue Ban, 2015 Mar;46(2):214-7.
2) Anesth Analg. 2014 Apr;118(4):745-54.
3) Br J Anaesth. 2013 Nov;111(5):825-32.
4) Bioorg Med Chem Lett. 2013 Mar 15;23(6):1813-6.
5) Eur J Mass Spectrom (Chichester). 2011;17(4):405-13.

It needs to provide sufficient rational reason for this study.

Experimental design

no comment

Validity of the findings

Because the previous report, couldn't found the impressive novelty.

Additional comments

1. To verifying the preclinical analysis, authors need to detect the metabolic enzymes such as p450 families with mRNA levels.
2. Figure 1 was showed several times in previous publication therefore don't need anymore here.
3. Author need to discuss in detail about differences between previous reported.

Reviewer 2 ·

Basic reporting

1. As it was described in article, the major objective of this study was to develop a new water-soluble pro-drug of propanol with non-toxic metabolites. (Line 223 & 224). Therefore, I think propofol should be taken as the major reference standard, when they discussed pharmacokinetics and pharmadynamics of the novel substance HX0969W, rather than fospropofol disodium. While surely when it comes to toxic metabolites, HX0969W should be compared with fospropofol disodium.
As we all know (and as the writer pointed in Line 248), propofol has been the most widely used intravenous anesthetic in clinical practices due to its rapid onset and recovery from sedation. Unfortunately, the novel substance HX0969W as a prodrug was deprived the characteristics.
It was described in the article like
“For onset time, propofol (0.4±0.1 min) had a shorter time to generate LoRR than HX0969W (1.8±0.4 min, p=175 0.017) and fospropofol disodium (2.1±0.7 min, p = 0.006). However, the duration of the LoRR 176 for HX0969W (75.8±9.6 min, p = 0.036) and fospropofol disodium (68.5±18.4 min, p= 0.041) 177 were significantly longer than that caused by propofol (27.1±6.0 min).” (Line 173-177)
“In this pharmacokinetic study, the longer onset time and duration of HX0969W and fospropofol disodium compared to propofol were owing to their pharmacokinetic features.” (Line 210-212)
“The time to achieve maximal concentration of propofol by HX0969W and fospropofol disodium was lengthened at 8 to 9 fold than propofol after intravenous administration. “ (Line 234)
In my opinion, the longer onset time and recovery time may not be well received by clinical anesthesiologists. It might not be highlight, but be a disadvantage.

2. As it was suggested in article, “Therefore, a slow-released propofol from HX0969W and fospropofol disodium reduced the risk of apnea, cardiovascular system depression (bradycardia, arterial blood pressure, systolic blood pressure, cardiac output, cardiac index, stroke volume index, and systemic vascular resistance), and convulsion (Line 244-247)”, which can be a satisfied reason to develop that a prodrug with longer onset and recovery time. Therefore, please provide solid support for the opinion.
It could be widely acceptable literatures (I haven't seen in manuscript), or an additional experiment to compare the effect of HX0969W and propofol on rat circulatory system, like blood pressure, QT interval, etc.

3. The data list in tables were not matched with figures, making the Cmax is obvious and really confusing. Please do not use mean±SD and mean±SEM alternatively, just use same format in a whole article.

Experimental design

4. Quantification of HX0969W, fospropofol disodium and propofol in rat plasma was necessary for pharmacokinetics study, thus the methodology work should be properly displayed.
At least typical chromatogram, linear regression of standard samples and precision and accuracy of quality control samples, etc.
For HX0969W and fospropofol disodium, they were analyzed via LC-MS/MS without stable isotope labeled internal standard, thus a matrix effect test could be taken into consideration.
For propofol, the focal point may be selectivity. Propofol would be or not be interfered by HX0969W and fospropofol in a HPLC-fluorescence assay, which should be clarified.
Methodology could be submitted as supplementary materials.

Validity of the findings

have no comments.

Additional comments

NA

Reviewer 3 ·

Basic reporting

No comment

Experimental design

The purpose of your experiment was to study the sedative-hypnotic effects of HX0969W, but did you consider the effect of pentobarbital sodium on your results at the end of the experiment.

Validity of the findings

Although HX0969W and fospropofol disodium had a longer time and longer duration to produce LoRR than propofol in rats, according to your pharmacokinetics, they take longer to work, how do you explain this.

Additional comments

You need to add more experiments to prove that HX0969W is superior to propofol in anesthetic effect

---

## Round 0.2 · accepted · Accept

Though no further experiments were supplemented as requested probably due to the outbreak of COVID-19, the authors have reasonably responded to the comments of the Reviewers.

Reviewer 1 ·

Basic reporting

Authors addressed the comments.

Experimental design

Authors addressed the comments.

Validity of the findings

Authors addressed the comments.

Additional comments

Authors addressed the comments.

Reviewer 2 ·

Basic reporting

I have reviewed the resived manuscript and all the supplementary material.
It is still a hypothesis that the pro-drug could show lower incidences of hypotension, but now I accept it is reasonable since the author has given some evidences for supporting their opinion.
I have no further comments.

Experimental design

no comment

Validity of the findings

no comment